# Grape Seed Proanthocyanidins Induce Autophagy and Modulate Survivin in HepG2 Cells and Inhibit Xenograft Tumor Growth in Vivo

**DOI:** 10.3390/nu11122983

**Published:** 2019-12-06

**Authors:** Lihua Wang, Weidong Huang, Jicheng Zhan

**Affiliations:** Beijing Key Laboratory of Viticulture and Enology, College of Food Science and Nutritional Engineering, China Agricultural University, Beijing 100083, China; lihuawang@cau.edu.cn (L.W.); huanggwd@263.net (W.H.)

**Keywords:** hepatocellular carcinoma, HepG2 cells, grape seed proanthocyanidins, autophagy, survivin, xenograft

## Abstract

Liver cancer is one of the leading causes of death worldwide. Although radiotherapy and chemotherapy are effective in general, they present various side effects, significantly limiting the curative effect. Increasing evidence has shown that the dietary intake of phytochemicals plays an essential role in the chemoprevention or chemotherapy of tumors. In this work, HepG2 cells and nude mice with HepG2-derived xenografts were treated with grape seed proanthocyanidins (GSPs). The results showed that GSPs induced autophagy, and inhibition of autophagy increased apoptosis in HepG2 cells. In addition, GSPs also reduced the expression of survivin. Moreover, survivin was involved in GSPs-induced apoptosis. GSPs at 100 mg/kg and 200 mg/kg significantly inhibited the growth of HepG2 cells in nude mice without causing observable toxicity and autophagy, while inducing the phosphorylation of mitogen-activated protein kinase (MAPK) pathway-associated proteins, p-JNK, p-ERK and p-p38 MAPK and reducing the expression of survivin. These results suggested that GSPs might be promising phytochemicals against liver cancer.

## 1. Introduction

Globally, liver cancer remains a serious threat to human health. Hepatocellular carcinoma (HCC) accounts for about 85–90% of all primary liver malignancies [1]. Most liver cancer patients are already in the advanced stage when they are diagnosed, unlike the early stage patients whose malignant tissue can be removed by surgical resection [2]. Although radiotherapy and chemotherapy are effective in general, they present various side effects, significantly limiting the curative effect [3].

Increasing evidence has shown that the dietary intake of proanthocyanidins plays an essential role in the chemoprevention or chemotherapy of tumors [4]. In vitro and in vivo toxicity experiments have demonstrated that proanthocyanidins are devoid of toxicity [5] and have an anticancer effect on various human cancers, such as colorectal cancer [6,7,8], pancreatic cancer [9], HCC [10,11], non-small cell lung cancer [12,13], squamous cell carcinoma [14], as well as head and neck squamous cancer [15].

Various fruits, beans, chocolates, fruit juices, wine, beer, and tea are rich in proanthocyanidins, and the proanthocyanidins in grape seeds are the most abundant [16]. Grape seed proanthocyanidins (GSPs) are formed by the polymerization of catechins and/or epicatechins, in the form of dimers, trimers, tetramers, and oligomers/polymers [17]. Although proanthocyanidins reportedly have anticancer effects on some human cancers, the precise mechanisms of HCC-associated cell death remain unclear. A previous study indicated that GSPs could markedly inhibit the growth of HepG2 liver cancer cells and induce apoptosis and the phosphorylation of mitogen-activated protein kinase (MAPK) pathway-associated proteins, p-JNK, p-ERK and p-p38 MAPK in vitro [18]. Therefore, this work further investigated whether GSPs can induce autophagy in HepG2 cells, its relationship to apoptosis, and key macromolecules involved, as well as GSPs effects on a HepG2-derived xenograft mouse model. This work provides supporting evidence for GSPs anti-cancer properties and position GSPs as promising phytochemicals against liver cancer.

## 2. Materials and Methods

### 2.1. Reagents and Antibodies

Dulbecco’s Modified Eagle Medium (DMEM, 4.5 g/L D-glucose, L-glutamine and 110 mg/L sodium pyruvate), 0.25% trypsin-EDTA and penicillin streptomycin were purchased from Gibco, Thermo Fisher Scientific, Inc. (Waltham, MA, USA). Fetal bovine serum was purchased from the Every Green, Zhejiang Tianhang Biotechnology Co., Ltd. (Hangzhou, China). GSPs (Purity ≥ 95.0%) were purchased from the Chengdu Must Bio-technology Co., Ltd. (Chengdu, Sichuan, China) and dissolved in DMSO during cell treatment and in drinking pure water (Hangzhou Wahaha Group Co., Ltd., Hangzhou, China) during xenograft treatment of nude mice. Antibodies against p-JNK (Thr183/Tyr185), p-ERK (Thr202/Tyr204), and p-p38 MAPK (Thr180/Tyr182) were purchased from Cell Signaling Technology (Boston, MA, USA). Antibodies against JNK, ERK, p38 MAPK, survivin, LC3, GAPDH, and Ki67 were purchased from Proteintech Group, Inc. (Wuhan, Hubei, China). 3-methyladenine (3-MA) was purchased from Sigma-Aldrich (St. Louis, MO, USA).

### 2.2. Cell Culture

The HCC HepG2 cells were generously gifted by Prof. Hongbo Hu (China Agricultural University, Beijing, China). HepG2 cells were cultured in DMEM supplemented with 10% heat-inactivated fetal bovine serum and 1% penicillin streptomycin. All cells were grown in a 5% CO_2_ humidified incubator at 37 °C.

### 2.3. RNA Isolation and Quantitative Real-Time Polymerase Chain Reaction (qPCR) Analysis

The HepG2 cells were grown in 6-well plates, and the total RNA was extracted using Trizol (Takara, Dalian, China). cDNA was prepared according to the manufacturer’s instructions using a HiFiScript cDNA Synthesis Kit (Cwbio, Beijing, China). The amplification of GAPDH was used as the internal reference gene to normalize the expression of the selected genes. The primer sequences were survivin-sense (5′-TACGCCTGTAATACCAGCAC-3′); survivin-antisense (5′-TCTCCGCAGTTTCCTCAA-3′) [19]; GAPDH-sense (5′-TCTGGTAAAGTGGATATTGTTG-3′); and GAPDH -antisense (5′-GATGGTGATGGGATTTCC-3′) [20].

Two-step qPCR was performed using a CFX96 Connect^TM^ Real-Time PCR System (Bio-Rad, USA). Each reaction was conducted in triplicate with a reaction volume of 20 μL containing 0.4 μL of each primer (10 μM), 10 μL of the UltraSYBR mixture (Cwbio, Beijing, China), 7 μL of diluted cDNA, and 2.2 μL of ddH_2_O. A thermal cycling protocol, involving pre-denaturation at 95 °C for 10 min, followed by 40 cycles of amplification (denaturation at 95 °C for 15 s and annealing at 60 °C for 1 min), was used. The melting curve analysis was conducted from 65 °C to 95 °C. Relative gene expression was calculated using the 2^−ΔΔCT^ method, as described by Livak & Schmittgen [21]. Untreated cells were considered to be the reference sample, which was defined as expression =1, and the results were expressed as the fold-change in comparison with the reference sample.

### 2.4. Apoptosis Analysis with Flow Cytometry

The apoptosis of HepG2 cells was analyzed with flow cytometry using the TransDetect Annexin V-FITC/PI Cell Apoptosis Detection Kit (Transgen Biotech, Beijing, China). Briefly, cells were plated into 6-well plates and cultured for 24 h. Following this treatment, the floating and adherent cells were collected, washed twice with cold PBS, and resuspended in 100 µL of ice-cold 1×Annexin V Binding buffer followed by a mixture of 5 µL of Annexin V-FITC and 5 µL of PI. The cells were incubated for 15 min in the dark at room temperature followed by mixing of 400 µL of ice-cold 1×Annexin V Binding buffer. The stained cells were then detected using a FACSCalibur flow cytometer (BD Biosciences, San Jose, USA). The data were analyzed by FlowJo 10 software (Tree Star, Inc., Ashland, OR, USA).

### 2.5. Western Blot Analysis

Briefly, cells were washed twice with cold PBS and then scraped off in a RIPA lysis buffer (50 mM Tris (pH 7.4), 150 mM NaCl, 1% Triton X-100, 1% sodium deoxycholate, 0.1% SDS, sodium orthovanadate, sodium fluoride, EDTA, and leupeptin) (Beyotime Biotechnology, Shanghai, China) containing protease inhibitor PMSF (1 mM). Tumor tissues were added to the RIPA lysis buffer containing protease inhibitor PMSF (1 mM), ground with a tissue grinder, and centrifuged to obtain the supernatant. The concentration of the protein samples was quantified using the BCA protein assay (Pierce^®^ BCA Protein Assay Kit, Thermo Fisher Scientific, MA, USA). Equal amounts of denatured proteins (20–40 µg/well) were subjected to SDS-PAGE gel (10% or 15%) electrophoresis and transferred onto polyvinylidene fluoride (PVDF) membranes (Immobilon^®^-P Transfer Membrane, Millipore) via wet transfer. The PVDF membranes were then blocked in 5% skim milk in TBS‑Tween‑20 (TBST) for 1 h at room temperature and incubated overnight with specific primary antibodies at 4 °C. Each membrane was washed three times with TBST and incubated with secondary antibody-horseradish peroxidase (HRP) conjugated with anti-rabbit IgG diluted in 5% skim milk at room temperature for 1 h, followed by three washes with TBST. Finally, immunoreactive bands were exposed to enhanced chemiluminescence (ECL) reagents to visualize the HRP signal.

### 2.6. Observation of Acidic Vesicular Organelles (AVOs) Formation

The observation of AVOs formation was performed as previously described using acridine orange (AO) staining [22]. Briefly, GSPs-treated cells were washed twice with PBS, followed by staining with 1 µg/mL AO for 30 min at room temperature. Afterward, cells were washed twice with PBS and observed using a fluorescence microscopy (Nikon Eclipse Ti) equipped with a green filter.

### 2.7. Cell Transfection

Cells were grown on 6-well cell culture plates for 24 h and then transfected with 2.5 µg of pQCXIP-GFP-LC3 for 24 h, followed by treatment with GSPs for 24 h. Following GSPs treatment, the formation of autophagic puncta was detected with a fluorescence microscope equipped with a blue filter.

For the transfection of pcDNA3.1-survivin, cells were grown on 6-well cell culture plates for 24 h and then transfected with 2.5 µg of pcDNA3.1-survivin for 24 h, followed by treatment with GSPs for 24 h. Afterward, apoptosis of the HepG2 cells was analyzed with flow cytometry.

### 2.8. HepG2-Derived Nude Mice Xenograft Model

Female BALB/C nude mice (4–5 weeks old) were purchased from the Beijing Vital River Laboratory Animal Technology Co., Ltd. (Beijing, China) and housed there in a SPF barrier environment that was maintained at a constant temperature (23–25 °C) and humidity (50–60%). All animals had free access to drinking water and food and received humane treatment. The animal protocol was approved by the Institutional Animal Care and Use Committee, Beijing Vital River Laboratory Animal Technology Co., Ltd.

HepG2 cells (2.5 × 10^6^ cells in 100 µL PBS per mouse) were injected subcutaneously on the right side of the back of nude mice. Thirteen days later, the tumor volume reached about 100 mm^3^, and mice were randomly divided into control (pure water) and GSPs treatment groups (100 mg/kg and 200 mg/kg body weight) via oral daily gavage. The length and width of the tumor were measured with a vernier caliper, and the mice were weighed every other day to determine their respective body weight. The volumes of the tumors were calculated using the formula: Volume = (length × width^2^)/2. At the termination of the experiment, the mice were sacrificed by CO_2_ euthanasia, and the tumor mass was harvested and weighed. A portion of the tumor tissue was paraffin-embedded for immunohistochemistry, and the other part was frozen in liquid nitrogen and stored at −80 °C for further analysis.

### 2.9. Immunohistochemistry

Ki67 staining was performed according to an immunohistochemical staining standard protocol. The samples were incubated overnight with ki67 antibodies (1:1000) at 4 °C. The HRP-labeled secondary antibody was then incubated at room temperature. Staining with 3,3′-diaminobenzidine was used as a chromogen and Mayer’s hematoxylin as a counterstain to the sections. Image-Pro Plus 6.0 (Media Cybernetics, Inc., Rockville, MD, USA) software was used to select the same brown-yellow cell nuclei as the unified standard for judging all photo-positive cells. The same blue cell nuclei as other cells were selected, and each photo was analyzed to obtain the number of positive cells and total cells, while the positive rate (%), positive cells / total cells × 100, was calculated.

### 2.10. Histopathological Examination

The paraffin sections of the tumor were deparaffinized in xylene and rehydrated through descending concentrations of ethanol according to routinely used methods and were stained with hematoxylin and eosin (HE) as previously described [23]. All sections were examined under a light microscope.

### 2.11. Statistical Analysis

The data of three independent experiments were expressed as mean ± standard deviation (SD). Statistical analysis was performed with an analysis of variance using SPSS version 21 (SPSS Inc., Chicago, USA). Duncan’s multiple range test was performed to determine the significant difference. Differences at *p* < 0.05 were considered to be statistically significant.

## 3. Results

### 3.1. GSPs Induced Autophagy in HepG2 Cells

The change in autophagy marker LC3 was first detected with Western blotting to investigate whether GSPs could induce autophagy in HepG2 cells. The expression of LC3 II is increased when autophagy occurs [24]. As shown in Figure 1a, the expression of LC3 II increased dramatically after treatment with 10 mg/L GSPs for 24 h and 48 h, respectively in HepG2 cells compared with the control group. Further confirmation regarding whether GSPs could induce autophagy in HepG2 cells was obtained by transfecting these cells with pQCXIP-GFP-LC3 for 24 h followed by treatment with 10 mg/L GSPs for 24 h to observe autophagic puncta. Figure 1b shows the formation of autophagic puncta (red arrow indication) in GSPs-treated cells transfected with pQCXIP-GFP-LC3 using a fluorescence microscopy. Also, to further demonstrate that GSPs treatment could induce autophagy in vitro, HepG2 cells were further stained with AO. AO is a fluorescent dye that crosses the cell membrane and enters the cell nucleus to form a uniform green fluorescence indicating DNA. AO can be protonated and trapped in AVOs, resulting in its metachromatic shift to red fluorescence [25]. Therefore, the fluorescence intensity of AO can directly reflect the number of autophagic vacuoles formed in the cells, that is, a higher fluorescence intensity causes the formation of more autophagic vacuoles. As shown in Figure 1c, the red fluorescence in HepG2 cells was markedly enhanced after GSPs treatment for 24 h and 48 h, confirming that GSPs could induce autophagy in HepG2 cells.

### 3.2. Inhibition of Autophagy Increased Early Stage Apoptosis of HepG2 Cells

Results indicated that GSPs could induce both apoptosis [18] and autophagy (Figure 1) in HepG2 cells. To investigate the relationship between apoptosis and autophagy, HepG2 cells were pretreated with the autophagy inhibitor, 3-MA (1 mM) for 1 h, and then treated with GSPs for 24 h, after which apoptosis was measured with flow cytometry (Figure 2). The results showed that cells in the early stage of apoptosis increased after the inhibition of autophagy, but no significant effect on the number of cells in the late stage of apoptosis was observed. These findings suggested that GSPs might cause the two forms of programmed death, apoptosis and autophagy, to cascade and transform, which constituted a complex system of programmed cell death together.

### 3.3. GSPs Significantly Reduced the Expression of Survivin in HepG2 Cells

Survivin plays an essential role in the regulation of apoptosis. Therefore, the changes in survivin at the mRNA and protein levels after treatment with GSPs were determined first. Results indicated that treatment with GSPs for 24 h and 48 h significantly reduced the expression of survivin at the mRNA and protein levels in HepG2 cells (Figure 3).

### 3.4. Survivin Was Involved in GSPs-induced Apoptosis

An overexpression vector of pcDNA3.1-survivin was constructed to further assess whether survivin was involved in GSPs-induced apoptosis. HepG2 cells were transfected with the overexpression vector for 24 h and then treated with 10 mg/L GSPs for 24 h. The apoptosis of the cells was measured with flow cytometry. The results showed that transfection of pcDNA3.1-survivin reduced the number of cells with early apoptosis induced by GSPs, but had no significant effect on the number of cells with late apoptosis (Figure 4).

### 3.5. GSPs Inhibited the Growth of HepG2 Cells without Displaying Observable Toxicity in Nude Mice

Since GSPs could significantly inhibit the growth of HepG2 cells in vitro [18], an investigation into whether GSPs could also inhibit the growth of HepG2 cells in vivo was conducted. The xenograft model in nude mice showed that 100 mg/kg and 200 mg/kg GSPs significantly reduced tumor size and tumor weight in nude mice (Figure 5a,b). The ki67 represented the degree of tumor cell proliferation, while the results of ki67 immunohistochemistry also showed that 200 mg/kg GSPs could significantly reduce the ki67 positive rate of the tumor (Figure 5c). Moreover, whether the dose of GSPs displayed observable toxicity in nude mice was also evaluated. Therefore, changes in the body weight of the nude mice during gavage with GSPs; two indicators of liver function, namely alanine aminotransferase (ALT) and aspartate aminotransferase (AST); an indicator of kidney function, creatinine (Cr); and HE staining of the liver were evaluated. The results showed that GSPs doses at 100 mg/kg and 200 mg/kg did not significantly affect body weight, ALT, AST, and Cr in nude mice, while the HE staining showed no damage to the liver of nude mice (Figure 5d–f).

### 3.6. GSPs Induced the Phosphorylation of the MAPK Pathway-Associated Proteins, and Decreased the Expression of Survivin

In the nude mouse xenograft model, GSPs enhanced the phosphorylation levels of the MAPK pathway proteins, and the most significant increase was displayed in the p-ERK level (Figure 6a). GSPs also reduced the expression of survivin in tumor tissues (Figure 6b). The results of Western blotting showed that GSPs did not enhance LC3 II in tumor tissues (Figure 6c).

## 4. Discussion

There is growing evidence that phytochemicals, which target the autophagy pathway, present a promising approach for cancer therapy [26]. The anti-tumor mechanism of these phytochemicals was verified to induce autophagy in cancer cells and cause programmed cell death [26]. Resveratrol, a non-flavonoid polyphenolic compound, induced autophagy in different ovarian cancer cell lines [27]. B-group triterpenoid saponins from soybean could induce macroautophagy, while down-regulating Akt and up-regulating ERK protein in human colon cancer cells [28]. Vitamin D analog EB1089 induced autophagy in MCF-7 breast cancer cells [29]. Pterostilbene induced autophagy on human oral cancer cells through the modulation of the Akt and MAPK pathways [22]. Procyanidins from *Vitis vinifera* seeds induced apoptotic and autophagic cell death via the generation of reactive oxygen species in squamous cell carcinoma cells [14]. Furthermore, in this work, the results also indicated that GSPs could induce significant autophagy in HepG2 cells (Figure 1), which may be one of the mechanisms by which GSPs inhibited the growth of HepG2 cells.

The in vitro studies have demonstrated that GSPs induced autophagy in HepG2 cells (Figure 1), while no significant autophagy was observed in the xenograft model (Figure 6c). This may be due to the following reasons. On the one hand, proanthocyanidins may exert an effect by interacting with other components of the gut, such as lipids or iron [30]. On the other hand, polymeric proanthocyanidins were catabolized by human colonic microflora into low molecular weight phenolic acids, such as *m*-hydroxyphenylpropionic acid, *m*-hydroxyphenylacetic acid, and *m*-hydroxybenzoic acid [31,32]. The bioavailability of GSPs and their metabolites in tumor tissues was also an important aspect [4]. Therefore, many factors, including studies in vitro, studies in vivo, and clinical trials in anti-tumor studies, should be considered. Further studies regarding the bioavailability, metabolism, toxicity, and pharmacodynamics of GSPs are necessary to promote the understanding of their beneficial effects on human health. The standardization of the dosage, composition of the grape seeds extract, and duration of the studies are also a necessity to shed light onto the cause-effect relationship between the intake of GSPs and their health effects in a more accurate way [33].

Survivin is the smallest member of the inhibitor of apoptosis (IAP) gene family [34,35] and not only inhibits apoptosis, promotes cell mitosis, and regulates the cell cycle, but also increases cell proliferation [36]. Studies have shown that survivin was highly expressed in most human tumor tissues, such as liver cancer, lung cancer, gastric cancer, pancreatic cancer, breast cancer, and other malignant tumors, but not in most normal tissues [37,38]. A phosphorothioate antisense oligonucleotide was the first described molecular antagonist of survivin, suppressing it in mRNA and protein expression, and producing strong anticancer activity in preclinical models [39]. In this work, results showed that GSPs significantly reduced the expression of survivin at the mRNA and protein levels (Figure 3). GSPs treatment significantly reduced the number of early apoptotic HepG2 cells transfected with the survivin overexpression vector (Figure 4). Survivin may be involved in GSPs-induced apoptosis, which is suggested as one of the pathways by which GSPs inhibit the proliferation of HepG2 cells.

Studies have also shown that GSPs were able to inhibit tumors in vivo. Previous studies showed that GSPs could inhibited the tumor growth of HeLa and SiHa cervical cancer cells [40], HCT116 colorectal cancer cells [41] and PC3 prostate cancer cells [42] in xenograft tumor model. Consistent with these studies, it was found that GSPs inhibited the growth of HCC, HepG2 cells in the xenograft of nude mice (Figure 5a–c). Moreover, it was found that proanthocyanidin-rich extract from grape seeds displayed a lack of toxicity during in vitro and in vivo toxicity experiments [5]. In this work, changes in the body weights of nude mice during gavage with GSPs, ALT and AST indicators of liver function, Cr indicator of kidney function, and the pathological HE staining of the liver were studied, indicating that the doses of GSPs at 100 mg/kg and 200 mg/kg caused no observable toxicity in nude mice (Figure 5d–f).

## 5. Conclusions

In conclusion, GSPs observably induced autophagy in vitro but not in vivo, and inhibition of autophagy increased the early stage apoptosis of HepG2 cells. In addition, GSPs modulated the expression of survivin, and survivin was involved in GSPs-induced apoptosis. In vivo studies showed that GSPs inhibited the growth of HepG2 cells without observable toxicity in nude mice; induced the phosphorylation of the MAPK pathway-associated proteins, p-JNK, p-ERK and p-p38 MAPK; and decreased the expression of survivin. Overall, the data suggest GSPs as promising phytochemicals with anti-cancer properties that can be potentially used to target HCC.

## Figures and Tables

**Figure 1 nutrients-11-02983-f001:**
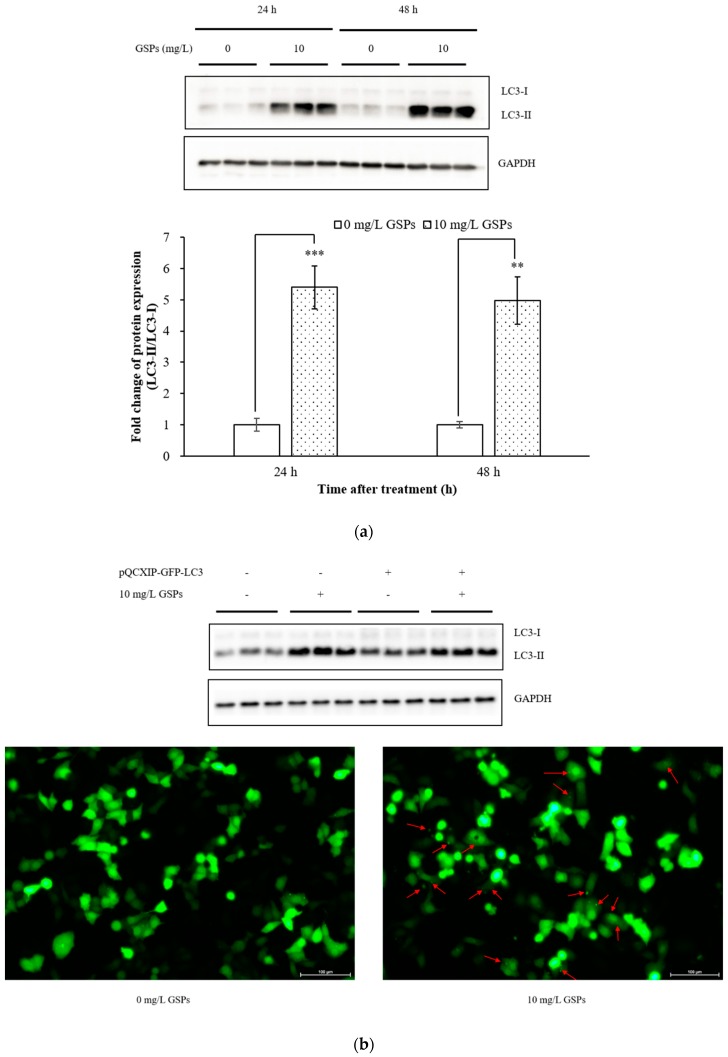
GSPs induced autophagy in HepG2 cells. (**a**) HepG2 cells were treated with 10 mg/L GSPs for 24 h and 48 h, and the protein expression of LC3-I and LC3-II was detected with Western blotting. (**b**) HepG2 cells were transfected with pQCXIP-GFP-LC3 for 24 h and then treated with 10 mg/L GSPs for 24 h. The transfection efficiency of pQCXIP-GFP-LC3 was detected with Western blotting, and the autophagic puncta (red arrow indication) were observed using a fluorescence microscope. (**c**) HepG2 cells were treated with 10 mg/L GSPs for 24 h and 48 h, then stained with AO (1 μg/mL), while AVOs formation was observed using a fluorescence microscope. The data of three independent experiments were expressed as mean ± SD. Duncan’s multiple range test was performed to determine the significant difference. ** and *** indicate that the values of treatment were significantly different at *p* < 0.01 and *p* < 0.001, respectively.

**Figure 2 nutrients-11-02983-f002:**
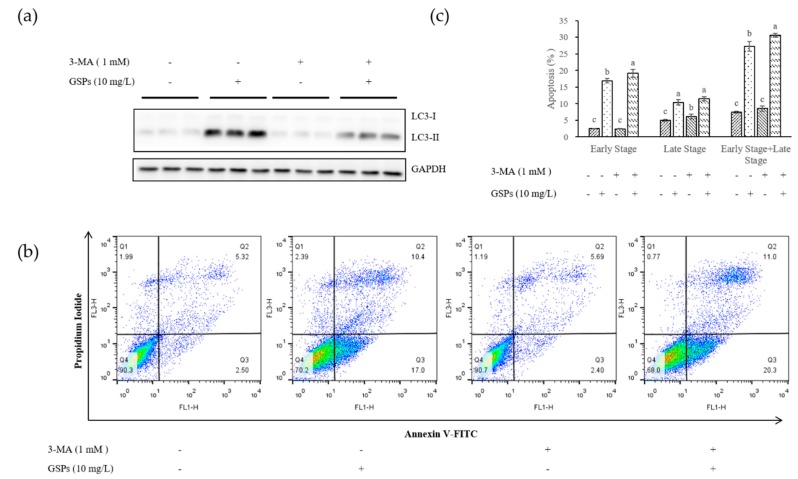
The inhibition of autophagy increased the apoptosis of HepG2 cells. HepG2 cells were pretreated with 3-MA (1 mM) for 1 h, then treated with GSPs (10 mg/L) for 24 h, and the protein was collected to determine the expression of LC3-I and LC3-II at the protein level using Western blotting (**a**), while the sample was collected to detect apoptosis with flow cytometry using Annexin V-FITC/PI (**b**). (**c**) Statistical plots of flow cytometry analysis for apoptotic cells. The data of three independent experiments were expressed as mean ± SD. Duncan’s multiple range test was performed to determine the significant difference. Different letters indicate significant differences at *p* < 0.05.

**Figure 3 nutrients-11-02983-f003:**
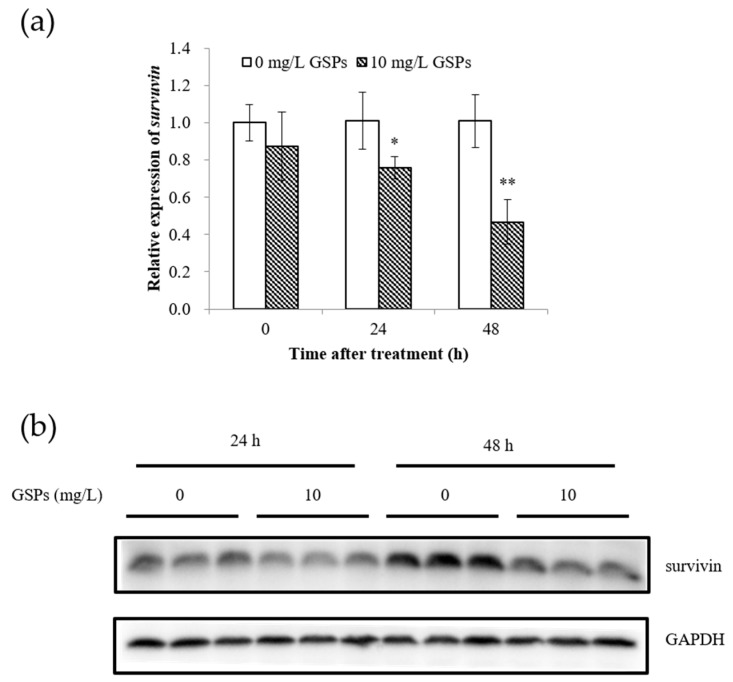
GSPs significantly reduced the expression of survivin. HepG2 cells were treated with 10 mg/L GSPs for 24 h and 48 h, while the expression of survivin was detected with qPCR at the mRNA level (**a**) and with Western blotting at the protein level (**b**). The data of three independent experiments were expressed as mean ± SD. Duncan’s multiple range test was performed to determine the significant difference. * and ** indicate that the values of treatment were significantly different at *p* < 0.05 and *p* < 0.01, respectively, compared with the untreated cell.

**Figure 4 nutrients-11-02983-f004:**
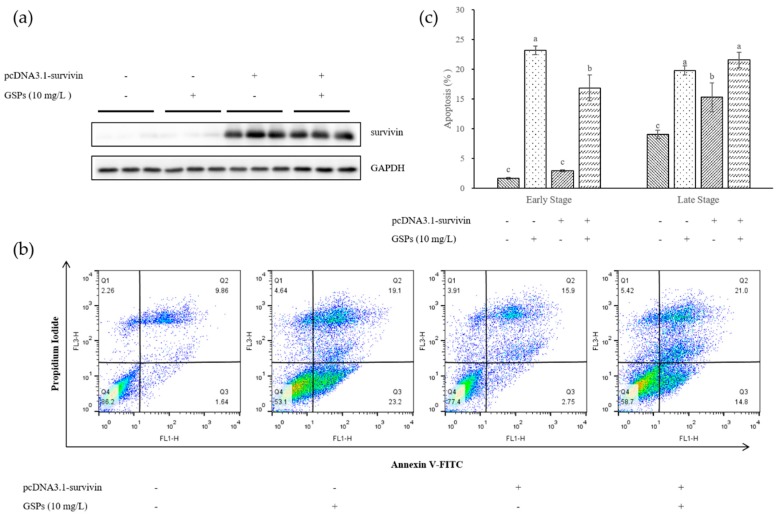
Survivin was involved in GSPs-induced apoptosis. The pcDNA3.1-survivin overexpression vector was transfected into HepG2 cells for 24 h. The HepG2 cells were then treated with 10 mg/L GSPs for 24 h, and the protein was collected to determine the transfection efficiency of pcDNA3.1-survivin using Western blotting (**a**), while the sample was collected to detect apoptosis with flow cytometry using Annexin V-FITC/PI (**b**). (**c**) Statistical plots of flow cytometry analysis for apoptotic cells. The data of three independent experiments were expressed as mean ± SD. Duncan’s multiple range test was performed to determine the significant difference. Different letters indicate significant differences at *p* < 0.05.

**Figure 5 nutrients-11-02983-f005:**
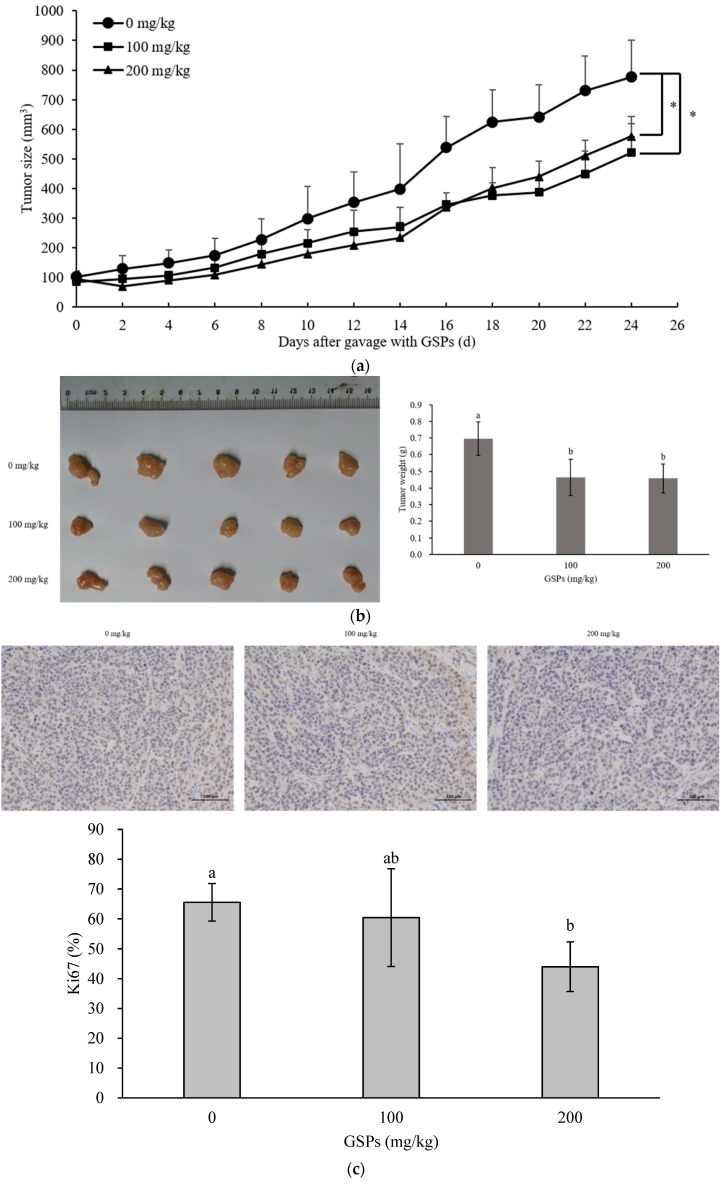
GSPs inhibited the growth of HepG2 cells without displaying observable toxicity in nude mouse model. (**a**) When the tumor size was about 100 mm^3^, the nude mice were treated with GSPs (0 mg/kg, 100 mg/kg, and 200 mg/kg) by oral gavage, while the length and width of the tumors were measured every other day with a vernier caliper, and the tumor volume (volume = (length × width^2^)/2) was calculated. (**b**) The mice were sacrificed after the GSPs treatment was concluded, and the tumors were removed and weighed. (**c**) Ki67 immunohistochemical staining was performed on the tumor tissues, and the statistics of ki67 positive rate were analyzed with Image-Pro Plus 6.0 software. (**d**) When the tumor size was about 100 mm^3^, the nude mice were treated with GSPs (0 mg/kg, 100 mg/kg, and 200 mg/kg) by oral gavage and were weighed every other day. (**e**) Following GSPs treatment, the nude mice were sacrificed, blood was collected from the heart, and the levels of alanine aminotransferase (ALT), aspartate aminotransferase (AST) and creatinine (Cr) in serum were determined. The *p* value is the difference between each group and the group of 0 mg/kg GSPs. (**f**) HE staining of the liver. The data were expressed as mean ± SD (*n* = 5). Duncan’s multiple range test was performed to determine the significant difference. * indicates that the values of the treatment are significantly different at *p* < 0.05 compared with the control. Different letters indicate significant differences at *p* < 0.05.

**Figure 6 nutrients-11-02983-f006:**
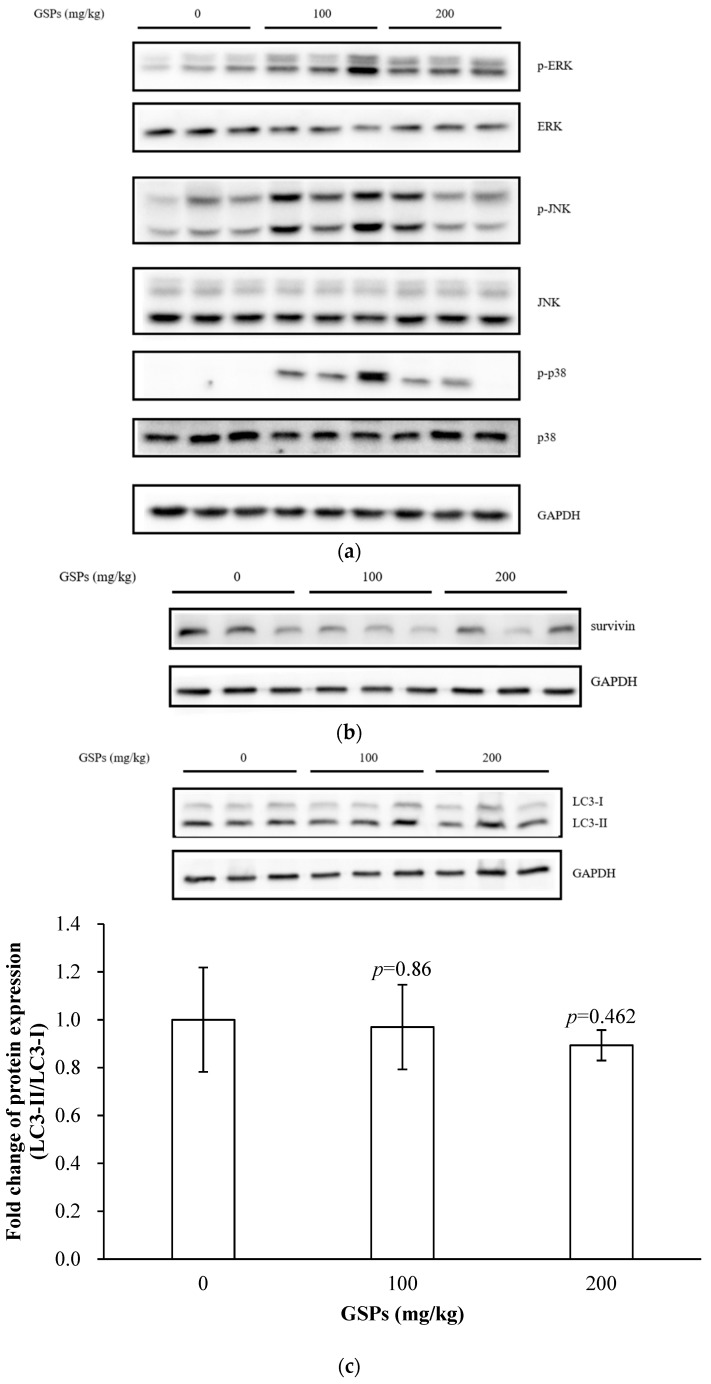
GSPs also induced the phosphorylation levels of the MAPK pathway proteins and decreased the expression of survivin but did not induce autophagy marker in tumor tissues. (**a**) Tumor tissues were lysed with RIPA lysis buffer, and the expression of the proteins involved in the MAPK pathway was detected with Western blotting. (**b**) The expression of survivin at the protein level was detected with Western blotting. (**c**) The expression of LC3-I and LC3-II at the protein level was detected with Western blotting. The data of three independent experiments were expressed as mean ± SD. Duncan’s multiple range test was performed to determine the significant difference. The *p* value is the difference between each group and the group of 0 mg/kg GSPs.

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
