# Peer review of "Grape Seed Proanthocyanidins Induce Autophagy and Modulate Survivin in HepG2 Cells and Inhibit Xenograft Tumor Growth in Vivo"

_nutrients, 2019, doi:10.3390/nu11122983_

Round 1

Reviewer 1 Report

Authors,

This is an  interesting study with potential in support of proanthocyanidins as anticancer agents. However, it is not very well executed, based on this write-up.

Overall the manuscript lacks clarity and needs substantial modifications including language and style. Additionally, the authors seem to oversell the outcome of this study, which can't be justified with the current work and manuscript.

Detailed comments:

Lines 2-5 -Title: Unsuited, based on the work presented. "Grape seed proanthocyanidins induced autophagy and regulated survivin in HepG2 cells in vitro and inhibited the growth of HepG2-derived xenografts in nude mice". The work does not support "gene regulation". HepG2 is indicated twice unnecessarily. - please rephrase

Lines 13-14:  Overstated facts. Please revise.

Line 18: What specific proteins within the MAPK pathway?

LIne 21: "phosphorylation of the MAPK " - inappropriate wording. Please rephrase!

Line 24: "HepG2-derived xenografts in nude mice". This is more of a phrase than a "keyword".

Line 28: "Discovered" inappropriate use of the word. Suggestion "they are diagnosed"

LIne 29: early stage "tumors"- replace with "malignant tissue"

Lines 29, 32, 39, 159, 303, 304: Reference(s) for the sentence?

Lines 3, 32, 42, 129, 245, 310, 312, 314, 351, 352, 357, 358, 360: Italicized word.

Line 34: "HCC cell" - repetitive, remove cell

Line 40, 60: "human HCC" and "HCC cell-associated"- repetitive, remove human and cell

Line 41: replace with -markedly. Significant is used with stats/numbers.

Line 41: "HepG2 (add- "liver cancer") cells (add-",") and induce apoptosis and the ("the"-remove)"

Line 42: "MAPK pathway" - complete name within parentheses

Line 45: "HepG2-derived xenografts in nude mice"- replace with- "HepG2-derived xenograft mouse model"

Line 51: "GSPs" compound- GSPs indicates a mixture/extract of different proanthocyanidins. If yes, please provide the exact components of the treatment. Each individual phytochemical should be listed. Include the catalog number and a web link to this specific product. (https://www.sciencedirect.com/science/article/pii/S0753332219305359)

Line 52: What is pure water? Please clarify.

Line 53:"Antibodies for use against", remove- "for use"

Line 61: "The HepG2 cells were cultured in DMEM medium", remove- "the" and "medium"

Line 61: High or low glucose DMEM?

Line 117, 119: "transfection of the pcDNA3.1-survivin," and "Afterward, the apoptosis", remove- "the"

Line 127: Injected where? Provide details of this model and/or provide references.

Line 129: "GSPs treatment"- If this is an extract/ combination of different phytochemical constituents: is the composition as same as the one used for in-vitro work?

Line 158: remove- "will be significantly" and add - "is"

Line 160: "10 mg/LGSPs" - add a space

Line 160:"and 48 h, respectively", remove ","

Line 163: too much use of -"whether"

Line 165: "as seen under", replace with "using"

Line 165: "microscope.", replace with "microscopy"

Line 166: "in HepG2 cells", remove, already established.

Line 166:"the HepG2", remove "the"

Line 167: "cell membrane-permeable", remove. It is explained in that sentence.

Line 168: "in combination with", replace with "indicating"

Lines 168-169: "AO can also bind to AVO", is this true? reference?

Line 173, 176: "in vitro", repetitive, remove.

Line 190-Figure 1: WB- Why is there no positive control?
Green fluorescence- The images are not clear and hard to distinguish. Quantify the fluorescent signal and include below the images.
AVO (image c) - Low quality. Having a nuclear stain would have helped to distinguish individual cells. Alternatively, a ratiometric analysis could have been done with AO, or could have used different dyes for autophagic vacuoles vs nucleus. See: https://www.ncbi.nlm.nih.gov/pubmed/27875278, https://www.ncbi.nlm.nih.gov/pubmed/24953340, https://www.ncbi.nlm.nih.gov/pmc/articles/PMC4341211/.

Line 198-Figure 2: Including a positive control (autophagy inducer) would have been appropriate and strongly supported the outcome.  Add letters to each individual figure such as (a),(b), etc

Line 205: Survivin -favor or disfavor apoptosis? reference

Lines 209-210: Sentence needs to be revised to include- MAPK pathway proteins, p38 MAPK, JNK, and ERK in HepG2 cells.

Lines 212, 213: "MAPK inhibitor"- What is this inhibitor?/remove/replace

Lines 213, 220, 312, 315: "Significantly" - To use this, the authors should quantify the WB band intensities and include a statistical/ percentage difference. Misuse of the word (out of context) "significantly". What percentage? Statistical value?

Line 214 -section 3.3- What about the observations/results of the combination of GSPs and MAPK inhibitors?

Line 220-Figure 3: A complete figure + description should be able to stand alone without the need for going back to the methods section. 3b includes 3 figures. Please separate the numbering/labeling. example: 3-a,b,c,d,e.

Line 222- "qPCR"- indicate the method used for calculation and normalizer.

Lines 227, 242: Stats method? please add.

Line 228: What is the control? un-treated

Line 237: separate figure labels - 4 a,b

Line 256: "results" - replace with staining

Line 256: "significant" - please remove. Describe the observations of the tissue architecture, etc.

Line 270-figure 5: Increase image size of b and f. separately label/ number the graphs in e. 

line 271: "nude mice" -replace with nude mouse model

Line 278: "weighed every other day" - Why was this done?

Line 280: "serum were determined" - for what?
Define ALT, AST, Cr.

Line 280: "HE staining of the liver" - for what and why?

Lines 283-284: "did not induce Autophagy in Vivo" -this implies the interpretation of the results, which should be done in the discussion. Please revise the sentence.

Line 2885: "xenograft model of the nude mice" -change to- nude mouse xenograft model

Line 286: "phosphorylation level of the MAPK pathway at the protein level,"- change to - levels of the MAPK pathway proteins 

Line 288: "in vivo"- replace with tumor tissue

Line 288-289: "meaning that no autophagy was observed." - no interpretation in this section.

Line 296- Figure 6: Why is the background signal intensity different from each other , especially the normalizer/control protein vs experimental protein? Were the phosphorylated and unphosphorylated blotting done on separate membranes?

Line 296: "phosphorylation of the MAPK pathway"- please revise. Inappropriate use of "MAPK pathway"

Line 297: autophagy or autophagy markers? Please revise

Line 298: "expression of the MAPK pathway at the protein level" - expression of the pathway or is it key proteins involved/ within the pathway? Please revise.

Line 299: LC3? what is it and why I and II? Better to include LC3II/LC3I relative intensity.

Line 304: What natural products? Please clarify

Line 313: What are the limitations of the in-vitro model? Can in-vitro concentrations be correlated to in-vivo concentrations?
What is the reported range for plasma concentrations in human? Why was HepG2 selected? Why weren't multiple cell lines were used for the in-vitro work ?, because cancer in general is a heterogeneous disease.

Line 315: "of nude mice" - please remove because it is implied by"xenograft model".

Line 316: remove - "the"

Lines 320-322: Does not make sense - Please revise and clarify. 

Line 325: "while no significant autophagy was observed in the xenograft model ". - insufficient justification (expand on the reasoning). 

Lines 330-332: How is this sentence/content relevant to this study? 

Line 333: Results are over interpreted without proper quantitation of protein.

Line 334: What can be said about the compound tested, based on this observation/result?

Lines 337-340: Sentence needs to be rewritten to include the investigated components of the MAPK pathway.

Line 340: "GSPs-induced apoptosis, which is suggested as one" -Are the authors suggesting this or is it from the literature?

Line 348: This section is merely a bunch of facts listed. Not completely integrated into the findings and conclusions of the outcome. Revise please.

Line 360: "regulate" - the results don't support gene regulation. Rather influence/have an impact/modulate. 

Lines 361-362: Please revise -"Overall, the data suggest GSPs as a promising phytochemical(s) with anti-cancer properties that can be potentially used to target HCC "

Reviewer 2 Report

OVERALL: There is previous evidence that GSE is a promising and effective treatment or prophylaxis against the development of human cancers. While at least one review of GSE effects on human cancer cell has omitted liver cancers (Dinicola, J Carc Mutagenesis, 2013), others have performed similar specific investigations into GSE apoptosis effects on in vitro liver CA cells (Hamza et al., Sci Rep, 2018). However, this manuscript appears unique in that it describes the effects and potential toxicity of GSP on HepG2 in vitro as well as HepG2-derived tumor graphs in vivo. This manuscript is exceptionally well-written and is very accessible to any reader of standard English familiar with the methods used. The methods & materials are very well described, and the conclusions are supported by the methods & sound results.

LINE 16: A comma after autophagy here is imperative for English readers. Please add.

LINE 18: What is this "w"? Is it italicized?

LINE 18: MAPK = Mitogen-activated protein kinase. This needs defined the first use.  Should read "..., while mitogen-activated protein kinase (MAPK) inhibitors..."

LINE 37: I suggest this phrase ("...especially grape seeds" be added as a separate sentence.

LINE 78: Perhaps add, "...method, as described by Livak & Schmittgen [17]."

LINE 88: (for readability) consider: "The stained cells were then detected using..."

LINE 153: I believe SPSS was acquired by IBM in 2009 or 2010. I believe you can keep "SPSS, Inc" but suggest adding city (i.e., "SPSS Inc, Chicago, USA)."

Figure 1: I struggle to see much on these graphs. I understand the analysis does not allow for enhancement of the images, but please ensure that the images are the best available.

LINE 302: Suggest "There is growing evidence that phytochemicals..."

LINE 338: Please change to "[14]"

LINE 363: suggest for Wang L.H. "..., performed the experiments, analyzed the data, and wrote the manuscript."

LINE 369: suggest changing to "...from the College of Food..." (i.e., adding "the")

References: Suggest adding DOIs to each reference.

LINE 410: I think the journals should not have periods after the abbreviations. i.e., this should read "J Sci Food Agric" (There are others as well....lines 378, 380, 382, etc.)

LINE 418: Please fix formatting.  

Reviewer 3 Report

This is an interesting manuscript that adds to the evidence that proanthocyanidins may have a preventive or therapeutic action against cancer. One of the difficulties in this field is that proanthocyanidins are a mixture of molecules that are likely to vary in different preparations. Another problem in elucidating the action of proanthocyanidins is that they appear to have multiple sites of action that may be primary or secondary effects. Some of these challenges have been described by Rodriguez-Perez et al. (Nutrients 11: 2435, 2019) and might be referenced in the manuscript.

            Additional words should be added to the sentence beginning on line 6 of the abstract. These words could be “the expression of survivin”. That would make the sentence like the heading of Section 3.3 in the Results and would prevent misinterpretation of the sentence in the Abstract to imply that GSPs increase MAPK inhibitors.

            In the Methods section 2.11, data are expressed as standard deviation (SD)/standard error of the mean (SEM). In the legends for Figures in the Results, SDs are presented for Figures 2, 3 and 4 but SEMs are given for Figure 5. The reason for the inconsistency is not clear. If the intention is to give an indication of variability or scatter, then SDs would be the better choice.

            In the Results, section 3.3, it is concluded that MAPK inhibitors significantly enhanced the expression of survivin at the protein level. Quantitation is not given for the results in Figure 3b and it is unclear from visual observation that the differences would be statistically significant. Quantitation of Western blots by densitometry has its problems but might be worth the effort in this case.

            Reference 14 cites the authors’ previous work and is in review. The title suggests there might be some overlap with the present manuscript. The authors might clarify if the present data are all novel or if there are confirmatory observations

Reviewer 4 Report

In this manuscript, Lihua Wang and co-authors examined the effect of Grape seed proanthocyanidins (GSP) on liver cancer. This study has some novelty since the results suggested that GSP increased the apoptosis in HepG2 cells and inhibited the growth of HepG2-derived xenografts in nude mice. However, the results from HepG2 cells and nude mice model are inconsistent regarding the effect of GSP on autophagy. Overall, the conclusion of this manuscript is not convincing because the results are not fully sufficient to support it. This manuscript needs some significant improvement to meet the criterion of publication. Several major concerns are listed as follows:

The authors need to apply a bigger scale in the following images: Fig1(b), (c) and Fig5(c), (f)

In the Fig1(b) and (c), it is necessary to present the images of related nuclear staining and the quantitation. Besides, it is really hard to tell that there is increased autophagy in cells with GSP treatment base on Fig1(b).

According to Fig3(a), the basal levels of survivin protein were significantly higher in 48h samples compared to 24h, while the mRNA levels didn’t reflect that. Could the authors provide some explanation on it? Overall, GSP decreased the surviving level more significantly at 48h time point. Did the authors use 48h as the time point to check the role of MAPK pathway in the following experiments?

In the line213, the authors said that “MAPK inhibitor significantly enhanced the expression of survivin at the protein level.” However, the blots in Fig3(b) didn’t support this conclusion. Besides, to confirm the inhibitor worked as expected, the authors should provide the level of phosphorylated proteins in the MAPK pathway, such as p38, ERK and JNK. The quantitation of the blots is also necessary since the changes are not obvious.

The image quality of Fig5(c) is very poor. Please replace it with a better one.

The authors need to provide some possible explanations about why the GSP didn’t induce any changes regarding autophagy.

Round 2

Reviewer 1 Report

Authors,

I appreciate the effort put into accepting and addressing the suggestions. The manuscript looks much better and it is much more clear to the reader. When there is no page limit or extra charge for additional pages it is always better to include all the details related to the experimental design as well as results. Additionally, the supplementary section can be always used to provide additional information. Including relevant controls, and reasoning behind experimental models and results  (including weight, concentration, gender, volume, etc) is part of a good manuscript based on the scientific process. This allows a wider audience to read and understand without the need to go back to the literature to find answers as well as gain interest. 

Please revise the following sentence (lines: 43-47) to highlight the objective and importance of the study.

"Therefore, this work further investigated whether GSPs can also induce autophagy, as well as the relationship between apoptosis in HepG2 cells, molecules involved in the modulation of apoptosis, and the effect of GSPs on the growth of HepG2-derived xenograft mouse model, which provides the evidence that GSPs might be promising phytochemicals against liver cancer."

Suggestion: Therefore, this work further investigated whether GSPs can induce autophagy in HepG2 cells, its relationship to apoptosis, and  key macromolecules involved, as well as GSPs effects on a HepG2-derived xenograft mouse model. This work provides supporting evidence for GSPs anti-cancer properties, and position GSPs as promising phytochemicals against liver cancer.

Best wishes!

Author Response

Dear reviewer,

Thank you very much for your attention. You are acknowledged from the bottom of our hearts for the careful reviews on our manuscript (nutrients-653845). According to your helpful comments, we make a careful revision on the original manuscript. All revisions are explained as follows:

I appreciate the effort put into accepting and addressing the suggestions. The manuscript looks much better and it is much more clear to the reader. When there is no page limit or extra charge for additional pages it is always better to include all the details related to the experimental design as well as results. Additionally, the supplementary section can be always used to provide additional information. Including relevant controls, and reasoning behind experimental models and results (including weight, concentration, gender, volume, etc) is part of a good manuscript based on the scientific process. This allows a wider audience to read and understand without the need to go back to the literature to find answers as well as gain interest. 

Point 1: Please revise the following sentence (lines: 43-47) to highlight the objective and importance of the study.

"Therefore, this work further investigated whether GSPs can also induce autophagy, as well as the relationship between apoptosis in HepG2 cells, molecules involved in the modulation of apoptosis, and the effect of GSPs on the growth of HepG2-derived xenograft mouse model, which provides the evidence that GSPs might be promising phytochemicals against liver cancer."

Suggestion: Therefore, this work further investigated whether GSPs can induce autophagy in HepG2 cells, its relationship to apoptosis, and key macromolecules involved, as well as GSPs effects on a HepG2-derived xenograft mouse model. This work provides supporting evidence for GSPs anti-cancer properties, and position GSPs as promising phytochemicals against liver cancer. 

Response 1: Thank you for your helpful suggestions. We have replaced with "Therefore, this work further investigated whether GSPs can induce autophagy in HepG2 cells, its relationship to apoptosis, and key macromolecules involved, as well as GSPs effects on a HepG2-derived xenograft mouse model. This work provides supporting evidence for GSPs anti-cancer properties, and position GSPs as promising phytochemicals against liver cancer" (lines 43-47) in revised manuscript.

The manuscript has been resubmitted to the journal. We look forward to your positive response.

Yours sincerely,

Jicheng Zhan

Reviewer 4 Report

The manuscript has been significantly improved. Most of the questions have been addressed in this revision. However, the scale bars were missing in several images such as Fig5c and f. Please add them properly.

Author Response

Dear reviewer,

Thank you very much for your attention. You are acknowledged from the bottom of our hearts for the careful reviews on our manuscript (nutrients-653845). According to your helpful comments, we make a careful revision on the original manuscript. All revisions are explained as follows:

Point 1: The manuscript has been significantly improved. Most of the questions have been addressed in this revision. However, the scale bars were missing in several images such as Fig5c and f. Please add them properly.

Response 1: Thank you for your helpful suggestion. We have added the scale bars in Figure 5c and f in revised manuscript.

The manuscript has been resubmitted to the journal. We look forward to your positive response.

Yours sincerely,

Jicheng Zhan